# Perceived safety and mental health among Asian American women: Exploring the moderating role of loneliness and resilience

Jiepin Cao[1]*, Sarah Shevrin[2], Linh M. An[3], Jennifer A. Wong[1], Sugy Choi[1], Stella S. Yi[1], Chau Trinh-Shevrin[1], Sahnah Lim[1]

**1** Department of Population Health, New York University Grossman School of Medicine, New York City, New York, United States of America, **2** College of Health Sciences and Technology, Rochester Institute of Technology, Rochester, New York, United States of America, **3** Hunter College AANAPISI Project, City University of New York, New York City, New York, United States of America

* jiepin.cao@nyulangone.org

## Abstract

Asian American women are an understudied population facing a substantial mental health burden, largely driven by the increasing rates of gender- and race-based violence and discrimination. Perceived safety, a key factor influencing mental health, has been underexplored. This study aimed to 1) examine the relationship between perceived safety with mental health outcomes (i.e., depression and anxiety); 2) examine the link between safety-related behavioral modifications and mental health outcomes; and 3) explore the moderating effects of loneliness and resilience on these relationships, drawing from a community sample of n = 345 Asian American women. Perceived safety was defined as feeling safe in public spaces, transportation and neighborhoods. In our study, 28.7%, 56.2% and 20.6% of women reported feeling unsafe in public spaces, transportation and neighborhoods, respectively. Feeling unsafe in public spaces was associated with higher odds of depression (aOR=2.37, 95%CI: 1.40, 4.02) and anxiety (aOR=2.61, 95%CI: 1.56, 4.37). Avoiding public spaces, leaving home or transportation were linked to increased odds of depression (aOR=2.05, 95%CI: 1.23, 3.40; aOR=2.26, 95%CI: 1.37, 3.74; aOR=2.03, 95%CI: 1.16, 3.57, respectively) while only avoiding leaving home was associated with anxiety (aOR=2.04, 95%CI: 1.24, 3.36). Loneliness moderated the association between avoiding public spaces and anxiety: avoiding public spaces was significantly associated with greater odds of anxiety among women who were not lonely (aOR = 4.77, 95% CI: 1.24–18.34), but not among those who were lonely (aOR = 0.93, 95% CI: 0.53–1.64). Resilience did not moderate these relationships (all $p > .050$). Our findings highlight the mental health impacts associated with reduced perceptions of safety and safety-related behavioral modifications in this group, highlighting the pervasive fear experienced by Asian American women in their day-to-day lives, and

**Data availability statement:** The data cannot be publicly shared due to ethical considerations. Access has been restricted in accordance with the New York University School of Medicine Institutional Review Board. Researchers who meet the criteria for accessing confidential data can contact NYULH IRB Operations: IRB-Info@NYULangone.org.

**Funding:** This work was supported by the National Institutes of Health (NIH) – National Institute on Minority Health and Health Disparities [grant number U54MD000538], NIH – National Heart, Lung, and Blood Institute [grant number 1R01HL160324], and NIH – National Institute of Mental Health [grant number R34MH136914-01A1].

**Competing interests:** The authors have declared that no competing interests exist.

emphasizing the need for targeted interventions to address the unique safety challenges faced by Asian American women.

## Intrvoduction

Asian Americans, a fast-growing minoritized population in the United States, are underrepresented in health research [1,2]. Existing literature suggests that this group, particularly Asian American women, faces a substantial burden of mental health challenges [3]. Emerging evidence since the onset of the COVID-19 pandemic indicates a significant worsening of depression and anxiety within this population, rising by 104% and 97%, respectively [4], and surpassing the rates observed in White Americans [5]. A major contributing factor to this rise is a dramatic 339% surge in anti-Asian hate crimes [6], which have continued to escalate over time. In December 2024 alone, anti-Asian hate crimes rose by 59% and have remained elevated [7]. A report from the Asian American Foundation reveals that half of the Asian American adults sampled in New York City have personally encountered hate incidents related to their race or ethnicity [8]. Asian American women are frequently the targets of these crimes, with 62% of such incidents affecting women [6]. This largely stems from their marginalization based on intersectional identities as both Asian and women, rooted in a legacy of gendered racism [3]. A recent national survey of Asian American, Native Hawaiian and Pacific Islander women revealed alarming trends across diverse Asian groups: 72% of East Asian, 75% of Southeast Asian, and 73% of South Asian women reported experiencing racism and/or discrimination in the past year; 35%, 37%, and 40% reported sexual harassment; and 9% of East and Southeast Asian women and 18% of South Asian women reported gender- or race-based physical violence [6]. Particularly concerning is that these experiences of racism and discrimination most commonly occurred in public spaces (e.g., restaurants and grocery stores), neighborhoods, and on public transportation. The tragic deaths of Michelle Go, who was pushed onto New York City subway tracks [9], and Yan Fang Wu, who was fatally shoved while walking home from a San Francisco bus stop [10], the brutal attack of 65-year old Vilma Kari [11], and the death of six Asian American women in Atlanta spa shooting in 2021 are some examples of the many and visible acts of violence in public spaces that have heightened the safety concerns among Asian American women. Beyond their direct victims, these incidents can have a profound impact through vicarious violence as individuals witness these incidents via social media, television, newspapers, or hear about them from others [12,13]. Consequently, 40% of Asian American women reported feeling less safe since the start of the pandemic [6], and 83% reported feeling unsafe in public spaces in New York City [8]; these feelings contribute to pervasive fear and avoidance behaviors [14], especially among women [8].

Perceived safety, a critical contributor to mental health outcomes, has been surprisingly overlooked in scholarly literature among Asian American women, especially considering the alarmingly high and rising rates of gender/race-based trauma

and violence in Asian American women. Gender differences are well-documented in the general population, with women expressing greater concern about personal safety and reporting feeling more vulnerable to threats than men [15–17]. Perceived safety, a critical but understudied social determinant of health [18], reflects the complex conditions of the social and built environments where people live, work, and play, including the balance of stressors and resources available to address those stressors [19]. Even without past victimization experiences, women in general may still feel fearful in public spaces, which can influence their social interactions, community engagement, and overall mental health. Strong evidence from existing studies has documented the negative effects of perceived safety on mental health in specific public settings, such as neighborhoods and college campuses, with links to increased depression and anxiety [20,21]. However, the broader implications of safety perceptions in various public areas that people rely on for their daily routines—such as restaurants, grocery stores, and public transportation—on mental health remain critically underexplored.

In response to perceived safety, individuals may frequently need to modify their behaviors, including adopting avoidance strategies, as a stress coping mechanism [22]. These behaviors involve alterations in daily routines to avoid specific places, people, or situations perceived as unsafe [23], such as staying away from certain areas or seeking companionship for a sense of security. For instance, Asian Americans who were fearful for their safety due to COVID-19 related racism and discrimination avoided taking public transportation, walking alone or exercising outdoors, grocery shopping, socializing with friends and family, and even missed in-person medical appointments [14,24]. Gender differences were also evident, with women generally more likely to engage in avoidance behaviors as a way of coping with safety concerns compared to men [15,24–26]. Emerging evidence suggests that these behavioral modifications may not effectively mitigate negative mental health impacts. Avoidant behaviors can reduce social interactions, diminish community engagement, and erode trust and cohesion, further harming their mental health [27]. However, empirical evidence on the influence of avoidant behaviors on mental health in Asian American women remains limited.

Loneliness, defined as the distressing experiences that arises from unmet social needs through inadequate support from their current social connections [28,29], is a global epidemic [30] with significant detrimental effects on mental health [31,32]. Current evidence on loneliness within Asian American communities is predominantly from older populations. Systematic reviews on elder Asian immigrants indicate that the prevalence of loneliness in this group is linked to reduced social interactions and small social networks [33], underscoring the need for increased community resources and care [34]. Loneliness can intensify the adverse effects of perceived safety threats and associated behavioral modifications on mental health. Asian Americans experiencing loneliness may tend to have restricted social networks [35,36] and limited social support from these networks [37], rendering them more susceptible to anxiety, depression, and other mental health issues [31,32] compared with their non-lonely counterparts, particularly when they feel unsafe or alter their behaviors in response to safety concerns. However, the influence of loneliness on the mental health of Asian American women, particularly in the context of environmental safety threats remains largely unexplored. Therefore, exploring the moderating effects of loneliness on the relationship between perceived safety, behavioral modifications, and mental health is essential for developing effective mental health interventions tailored to the unique needs of Asian American women.

Despite these challenges, Asian American women have demonstrated remarkable resilience. Resilience, characterized by the ability to bounce back from adversities and cope with challenging situations [38], is closely linked to positive mental health outcomes [39–41]. The mechanisms through which resilience buffers the impact of stress, such as perceived safety and safety-related behavioral modifications, on mental health are complex and likely involve the interplay of biological, psychological, social, and environmental factors [42]. Importantly, resilience is influenced by culturally relevant resources available in one's social and built environments [42,43]. Resilience can influence how individuals assess and respond to stressors [44]. Evidence from the general population indicates that women with high resilience are more likely to employ adaptive coping strategies than those with low resilience, which may better equip them to manage the stress associated with perceived safety and behavioral modifications [45]. In contrast, those with lower resilience may rely on passive coping strategies, such as social disengagement, to deal with stress, which is associated with decreased perceived control [46]

and increased vulnerability to stressors, thereby heightening their vulnerability to depression and anxiety. Understanding the potential moderating role of resilience is essential for identifying Asian American women who are at higher odds for mental health challenges; however, empirical evidence in this area remains limited. This understanding can guide the development of targeted interventions aimed at enhancing coping abilities and mitigating the negative effects of perceived safety.

To address these gaps in current literature, the research aims of the current study are to: 1) examine the relationships between perceived safety in public spaces, public transportation, and neighborhoods with mental health outcomes (i.e., depression and anxiety); 2) examine the associations between safety-related behavioral modifications (i.e., behavioral modifications in response to safety concerns) and mental health outcomes (depression and anxiety); 3) and to explore the moderating role of loneliness and resilience on these relationships among community-dwelling Asian American women.

## Materials and methods

### Design and procedures

This cross-sectional online study included a community sample of 345 participants who met the following eligibility criteria: (1) self-identified as an Asian or Asian American woman, (2) were 18 years of age or older, and (3) resided or stayed in the US at the time of the survey. The survey was administered using REDCap (Research Electronic Data Capture) electronic data capture tools hosted at New York University (NYU) Langone Health. REDCap is a secure, HIPAA (Health Insurance Portability and Accountability Act) compliant software platform designed to support data collection for research studies [47,48]. The study was approved by NYU Langone Health Institutional Review Board.

Participants were recruited from August 15th 2022 to June 22nd 2023. To reach potential participants, study flyers that detailed the purpose of the study and eligibility criteria were disseminated via social media platforms such as Instagram and email listservs of organizations dedicated to research or services for the Asian American communities. These organizations included the Center for the Study of Asian American Health at NYU, the Chinese American Family Alliance for Mental Health, and the Asian American Psychological Association. Interested individuals could access the screening form either by clicking on the Open REDCap link or by scanning the QR code provided in the flyer. Eligible individuals were invited to complete the informed consent electronically by clicking the "Next" button and subsequently proceeding to a self-administered online survey, which took approximately 30 minutes to complete. The survey was administered in English, and collected perceived safety, behavior changes and health-related information, and demographics. Following survey completion, respondents were offered a list of culturally appropriate mental health resources.

### Measures

**Depression.** The 9-item Patient Health Questionnaire (PHQ-9) was used to measure depressive symptoms over the past two weeks [49]. Participants rated the frequency that they were bothered by the symptoms on a scale from 0 ("*Not at all*") to 3 ("*Nearly every day*"). Example items include "Little interest or pleasure in doing things," "Trouble falling asleep, staying asleep, or sleeping too much," and "Feeling bad about yourself—or that you're a failure or have let yourself or your family down". The PHQ-9 has demonstrated validity and reliability in Asian American populations [50,51] and showed good internal consistency in our sample (Cronbach's $\alpha = .89$). To maximize the overall sensitivity and specificity of PHQ-9 to detect major depression [52], a dichotomized outcome was created based on the total score using a cut-off score of 10: 0 = minimal/mild depression (<10); 1 = moderate/moderately severe/severe depression (≥ [10]).

**Anxiety.** The Generalized Anxiety Disorder 7-item Scale (GAD-7) was used to assess anxiety over the past two weeks [53]. Participants rated how much they were bothered by a set of symptoms on a scale from 0 ("*Not at all*") to 3 ("*Nearly every day*"). Example items include "Feeling nervous, anxious or on edge" and "Worrying too much about different things". The GAD-7 has been found to have good reliability and validity among Asian Americans [54]. In our sample, this scale

demonstrated excellent internal consistency: Cronbach's α = .91. Participants were considered positive for anxiety if their total score of GAD-7 was greater or equal to 10 [53]: 0 = minimal/mild anxiety (<10); 1 = moderate/moderately severe/severe anxiety (≥ [10]).

**Perceived safety.** Participants were asked to rate their perceived level of safety in public spaces, on public transportation, and in their neighborhood on a 4-point scale from 1 ("*very safe*") to 4 ("*very unsafe*") over the past year, assessed through 4 items. These questions were motivated by the 2022 National Asian Pacific American Women's Forum report, which highlighted that public spaces, mass transit, and neighborhoods were the most common locations where Asian American and Pacific Islander (AAPI) women experienced gender and/or race-based discrimination, or violence [55]. For the current study, responses were dichotomized: 0 = safe (*very safe*); 1 = unsafe (all other responses than "*very safe*").

**Safety-related behavioral modifications.** Participants who did not perceive public spaces, public transportation, or neighborhoods as "*very safe*" were further asked whether they had ever changed their behaviors due to safety concerns (yes/no). This decision was based on our study aim to examine behavioral modifications specifically in response to perceived safety threats. Behavioral modifications included: staying away from public spaces, avoiding leaving their home, decreasing or avoiding the use of public transportation, and having someone accompany them when going outside [55]. Each behavioral modification was dichotomized: 0 = no; 1 = yes.

**Loneliness.** The UCLA 3-item loneliness scale, one of most commonly used scale for loneliness [56], was administered to measure loneliness [57]. Participants rated how often they felt a lack of companionship, felt left out, and felt isolated from others on a scale from 1 ("*Hardly ever*") to 3 ("*Often*"). This scale showed good internal consistency in current sample (Cronbach's α = .85). Although there is no consensus on a cut-off score for the total loneliness score, consistent with other studies, we used a total score greater than 6 to indicate the presence of loneliness [58]. Responses were dichotomized: 0 = without loneliness (<6); 1 = loneliness (≥ [6]).

**Resilience.** The Brief Resilience Scale was used to evaluate participants about their ability to recover when encountering adversities [38]. This scale has demonstrated good validity and reliability in Asian populations [59,60]. The internal consistency of this scale in our sample was good (Cronbach's α = .87). Participants were asked about the extent to which they agree with the 6 statements from "*strongly disagree*" to "*strongly agree*". Exemplary statements include "I have a hard time making it through stressful events" and "It does not take me long to recover from a stressful event". A higher score represents higher resilience. Following the scoring and categorization of the scale developer [38], resilience was dichotomized: 0 = low resilience (<3); 1 = normal to high resilience (≥ [3]).

**Sociodemographic information** including age, nativity (US-born: yes/no), years in the US, Asian ethnic groups (*What is your ancestry or ethnic identity? Please check all that apply [multiple responses allowed]*: Bangladeshi, Burmese, Bhutanese, Cambodian, Chinese, Filipino, Hmong, Indian, Indonesian, Japanese, Korean, Laotian, Malaysian, Mongolian, Nepali, Okinawan, Pakistani, Sri Lankan, Taiwanese, Thai, Vietnamese, or Other [please specify]), sexual orientation (straight/lesbian or gay/bisexual/other), and language spoken at home (English/Asian language/English and another Asian language equally/Other [please specify]), education (never attended school or only attended kindergarten/grades 1–8/grades 9–11/grade 12 or general educational development/college 1–3 years/college 4 years or more) and employment (yes, part-time [<35 hours per week]/yes, full-time [>35 hours per week]/no, not employed) were also collected. These variables were pre-selected based on established literature on their relevance to mental health outcomes of interests (i.e., depression and anxiety) among Asian American populations [61–63].

## Statistical analyses

Descriptive statistics were used to summarize sample characteristics, mental health correlates (perceived safety and safety-related behavioral modifications), moderators (loneliness, resilience) and mental health outcomes (depression and anxiety). Means, standard deviations and ranges were reported for continuous variables, while frequencies and

percentages were reported for categorical variables. All statistical tests were two-tailed, with a significance level set at $p = 0.05$. All analyses were performed using SAS 9.4 (Cary, NC).

Bivariate analyses were conducted to examine the relationships between pre-selected list of demographic variables (i.e., age, sexual orientation, employment, and education) and each mental health outcome to identify correlates to be included in multivariate logistic models.

Multivariable logistic regression analyses were used to assess the relationships between each mental health correlate and moderator with each mental health outcome. All correlates significant at the bivariate level were included in each model. Effect sizes were denoted by Adjusted Odds ratios (aORs) and their 95% confidence intervals (CIs).

To explore potential moderation effects, a two-way interaction term between each mental health correlate and the moderator was added to the multivariable logistic regression models, alongside the covariates, regardless of the significance of main effects of the mental health correlate and moderator. An interaction term was considered significant if the *p*-value in the Wald test was less than 0.05. aORs and 95% CIs were derived for variables involved in significant interactions. Additionally, interaction effects were visualized by plotting the two categorical variables against the outcome to help with the interpretation.

## Results

### Sample characteristics

In our sample of Asian American women, the average age was $34.8 \pm 9.1$ years old and had lived in the US for $8.7 \pm 13.7$ years. Majority of them self-identified as East Asian (n = 224, 64.93%), with the three largest ethnic groups being Chinese (n = 112, 32.1%), Korean (n = 48, 13.8%), and Filipina (n = 48, 13.8%). Most women identified as heterosexual/straight (76.5%), had at least a college education (92.8%) and were either full-time or part-time employed (87.4%). Additionally, a significant portion were born in the US (65.3%) and predominantly spoke English at home (70.4%). See Table 1.

### Perceived safety and mental health

Among our sample, 28.7% perceived public spaces as unsafe, 56.2% perceived transportation unsafe, and 20.6% perceived their neighborhoods as unsafe (See Table 1). Additionally, over one-fourth experienced mental health challenges, with 31.2% reporting moderate to severe depression and 27.5% reporting moderate to severe anxiety. Women who perceived the public space as unsafe were at a significantly higher odds of depression (aOR = 2.37, 95%CI: 1.40, 4.02) and anxiety (aOR = 2.61, 95%CI: 1.56, 4.37) compared to those who perceived public space to be safe.

### Behavior changes and mental health

Among Asian American women who perceived public spaces, transportation, or their neighborhood as unsafe (n = 342), nearly half modified their behaviors due to safety concerns (not mutually exclusive): 54.4% avoided public spaces, 42.4% refrained from leaving home, 65.7% avoided using transportation, and 46.5% ensured they had someone with them when outside (See Table 1). Women who stayed away from public spaces had 2.05 times the odds of experiencing depression compared to those who did not (aOR=2.05, 95%CI: 1.23, 3.40). Those who avoided leaving their homes for safety concerns had a 2.26 times higher likelihood of depression (aOR=2.26, 95%CI: 1.37, 3.74) and a 2.04 times higher likelihood of anxiety (aOR=2.04, 95%CI: 1.24, 3.36) compared to those who did not. Additionally, women who avoided public transportation due to safety concerns had 2.03 times the odds of reporting depression compared to those who did not (aOR=2.03, 95%CI: 1.16, 3.57). See Table 2.

### Moderation effects of loneliness

As shown in Table 2, women who reported loneliness had 4.71 times the odds of experiencing depression (aOR=4.71, 95%CI: 2.47, 8.99) and 3.71 times the odds of experiencing anxiety compared to those who did not report loneliness

**Table 1. Sample Characteristics of Asian American Women (n = 349).**

| | Mean±SD/N (%) |
|---|---|
| **Sociodemographic information** | |
| Age | 34.8±9.1, range: 19–70 |
| Born in the US | 228 (65.3) |
| Years in the US | 8.7±13.7, range: 0–51 |
| Asian American groups | |
| *Chinese* | 112 (32.1) |
| *Korean* | 48 (13.8) |
| *Filipina* | 48 (13.8) |
| *Taiwanese* | 31 (8.9) |
| *Vietnamese* | 27 (7.7) |
| *Indian* | 23 (6.6) |
| *Japanese* | 18 (5.2) |
| *Hmong* | 13 (3.7) |
| *Other\** | 29 (8.3) |
| Sexual orientation | |
| *Heterosexual/Straight* | 263 (76.5) |
| *Lesbian/Gay/Bisexual/Other* | 81 (23.5) |
| Education | |
| *Grade 12/GED/College 1–3 years* | 25 (7.2) |
| *>=4 years of college* | 324 (92.8) |
| Employed *(full time/part-time)* | 304 (87.4) |
| Language spoke at home | |
| *English* | 245 (70.4) |
| *Asian/both equally* | 103 (29.6) |
| **Perceived safety** | |
| Public space | |
| *Yes* | 249 (71.3) |
| *No* | 100 (28.7) |
| Public transportation | |
| *Yes* | 153 (43.8) |
| *No* | 196 (56.2) |
| Neighborhood | |
| *Yes* | 277 (79.4) |
| *No* | 72 (20.6) |
| **Safety-related behavioral modifications** | |
| Stay away from public spaces | |
| *Yes* | 186 (54.4) |
| *No* | 156 (45.6) |
| Avoid leaving home | |
| *Yes* | 144 (42.4) |
| *No* | 196 (57.6) |
| Avoid transportation | |
| *Yes* | 213 (65.7) |
| *No* | 111 (34.3) |

*(Continued)*

**Table 1.** (Continued)

| | Mean±SD/N (%) |
|---|---|
| Accompany | |
| *Yes* | 158 (46.5) |
| *No* | 182 (53.5) |
| **Moderators** | |
| Loneliness | |
| *Yes* | 229 (65.8) |
| *No* | 119 (34.2) |
| Resilience | |
| *Normal/high resilience* | 194 (55.6) |
| *Low resilience* | 155 (44.4) |
| **Mental health outcomes** | |
| Depression | |
| *Moderate/Moderately Severe/Severe Depression* | 111 (31.8) |
| *Minimal/Mild Depression* | 238 (68.2) |
| Anxiety | 29 (43.3) |
| *Moderate/Severe Anxiety* | 96 (27.5) |
| *Minimal/Mild Anxiety* | 253 (72.5) |

*Note.* * Other Asian" includes Bangladeshi (n=5), Thai (n=5), Indonesian (n=3), Malaysian (n=3), Pakistani (n=3), Nepali (n=2), Laotian (n=1), Mongolian (n=1), Okinawan (n=1), Sri Lankan (n=1), and participants who selected "Other" but did not provide a specific ethnicity (n=4). Asian subgroup was assessed using a multiple-choice question: "What is your ancestry or ethnic identity? Please check all that apply." Because participants could select more than one subgroup, the total percentage exceeds 100%.

**Table 2.** Adjusted Logistic Regression Models for Correlates of Mental Health Outcomes (n=345).

| | n (%) | Depression | | | Anxiety | | |
|---|---|---|---|---|---|---|---|
| | | aOR | 95% CI | p-value | aOR | 95% CI | p-value |
| Feel unsafe in public space (*ref=safe*) | 100 (28.7) | 2.37 | 1.40, 4.02 | **.001** | 2.61 | 1.56, 4.37 | **.001** |
| Feel unsafe in public transportation (*ref=safe*) | 196 (56.2) | 1.33 | 0.81, 2.19 | .267 | 1.22 | 0.74, 2.00 | .433 |
| Feel unsafe in neighborhood (*ref=safe*) | 72 (20.6) | 1.67 | 0.93, 3.00 | .086 | 2.46 | 1.40, 4.31 | **.002** |
| Stay away from public spaces (*ref=No*) | 186 (54.4) | 2.05 | 1.23, 3.40 | **.006** | 1.36 | 0.83, 2.23 | .227 |
| Avoid leaving home (*ref=No*) | 144 (42.4) | 2.26 | 1.37, 3.74 | **.001** | 2.04 | 1.24, 3.36 | **.005** |
| Avoid transportation (*ref=No*) | 213 (65.7) | 2.03 | 1.16, 3.57 | **.014** | 1.32 | 0.77, 2.26 | .319 |
| Accompany (*ref=No*) | 158 (46.5) | 1.15 | 0.69, 1.91 | .588 | 1.14 | 0.69, 1.88 | .607 |
| Loneliness (*ref=without loneliness*) | 229 (65.8) | 4.71 | 2.47, 8.99 | **.001** | 3.71 | 1.97, 6.97 | **.001** |
| Low Resilience (*ref=Normal/high resilience*) | 155 (44.4) | 3.26 | 1.96, 5.42 | **.001** | 3.82 | 2.28, 6.40 | **.001** |

*Note.* aOR = adjusted odds ratio; CI=Confidence interval for aOR; ref=reference group. For all models with the outcome depression, we controlled for age, sexual orientation, education, employment, and language spoken at home; For all models with the outcome anxiety, we controlled for age, sexual orientation, employment, and language spoken at home.

(aOR=3.71, 95%CI: 1.97, 6.97). Loneliness did not moderate the relationship between any safety-related behavioral modifications and depression (all *p* > .05). However, it moderated the relationship between staying away from public spaces and anxiety (See Table 3 and Fig 1). The Wald chi-square statistic for this interaction term was 4.81 (*p* = .028), indicating a significant moderation effect. Specifically, among women who did not feel lonely, those who stayed away from public

**Table 3. Moderation Effects of Loneliness on Depression and Anxiety.**

| Interaction term | Depression | | Anxiety | |
|---|---|---|---|---|
| | Wald Chi-square statistics | p-value | Wald Chi-square statistics | p-value |
| Feel unsafe in public space * Loneliness | 1.49 | .222 | 0.34 | .559 |
| Feel unsafe in public transportation* Loneliness | 0.88 | .349 | 0.86 | .355 |
| Feel unsafe in neighborhood * Loneliness | 1.57 | .210 | 0.36 | .549 |
| Stay away from public spaces * Loneliness | 2.31 | .128 | 4.81 | **.028** |
| Avoid leaving home * Loneliness | 0.65 | .421 | 2.43 | .119 |
| Avoid transportation * Loneliness | 1.46 | .227 | 2.66 | .103 |
| Accompany * Loneliness | 2.71 | .099 | 2.88 | .090 |

*Note.* Wald chi-square test statistics were for interaction terms from logistic regression models. *p* values are two-tailed. Each row represents a separate model testing the interaction between resilience and the listed safety/behavior variable in predicting depression or anxiety. Loneliness was dichotomized into without loneliness (≤6) and loneliness (>6).

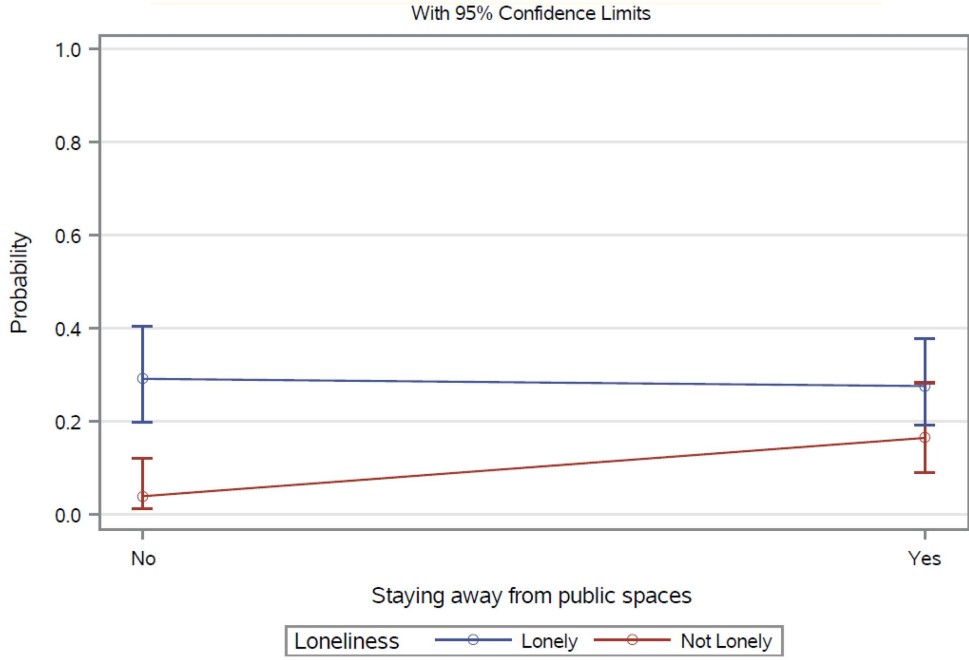

**Fig 1. Anxiety: interactions between stay away from public spaces and loneliness.** Note: This figure illustrates the moderation effect of loneliness on the association between staying away from public spaces and anxiety. The Wald chi-square statistic for the interaction term was 4.81 (p = 0.028). Conditional odds ratios from the simple slope analyses (aOR = 4.77 for non-lonely women; aOR = 0.93 for lonely women) are reported in-text but not displayed in this figure.

spaces due to safety concerns had 4.77 times the odds of reporting anxiety compared to those who did not stay away (aOR=4.77, 95% CI: 1.24, 18.34). However, among those who felt lonely, staying away from public spaces due to safety concerns was associated with lower odds of reporting anxiety compared to those who did not stay away (aOR=0.93, 95% CI: 0.53, 1.64). These conditional effect estimates are not presented in Table 3 or Fig 1 but are reported here to clarify the nature of the significant interaction.

## Moderation effects of resilience

As shown in Table 2, women who reported low resilience had 3.26 times the odds of experiencing depression (aOR=3.26, 95%CI: 1.96, 5.42) and 3.82 times the odds of experiencing anxiety (aOR=3.82, 95%CI: 2.28, 6.40) compared to those with normal to high resilience. The effects of perceived safety or behavior change due to safety concerns on depression did not differ by resilience status (all $p > .05$). Similarly, the impact of perceived safety or safety-related behavioral modifications on anxiety did not vary based on resilience status (all $p > .05$). See Table 4.

## Discussion

The current study contributes to the limited body of scientific literature on the role of perceived safety on mental health outcomes among Asian American women. To our knowledge, this study is among the first to examine how perceived safety in public spaces and related behavior modifications are linked to mental health outcomes within this understudied yet vulnerable population. Our findings indicate that feeling unsafe in public spaces and neighborhoods, alongside safety-related behavioral modifications such as avoiding leaving home or using public transportation, are significantly associated with increased odds of adverse mental health outcomes. Moreover, loneliness emerged as an important factor in moderating some of these relationships, although not all findings reached statistical significance. Collectively, these results highlight the importance of enhancing safety in public environments beyond neighborhoods and their implications for mental health within this vulnerable population.

Our study underscores the alarmingly high rates at which Asian American women perceive a lack of safety in public sites. This is often linked to the prevalent race and/or gender-based discrimination and harassment against this group, as evidenced by a recent community-based survey of Asian American, Native Hawaiian and Pacific Islander women (n=2,414) reporting that 74% of them experienced racism or discrimination within the past 12 months, most commonly in public spaces, transportation, and neighborhoods [6]. While data specifically documenting the perceived safety of the public areas among this population is limited, emerging research from Asian Americans during the COVID-19 and Asian American college students indicate similarly concerning trends. As many as 78% of Asian Americans in NYC reported feeling fearful for their safety because of racism or discrimination related to the COVID-19 pandemic [14]. Additionally, between 25% and 52% Asian American college students reported feeling unsafe on campus or in surrounding areas at night [64], potentially driven by fears of sexual assault and racism [65,66]. Moreover, gender differences were observed where women consistently report higher rates of lack of perceived safety and suffer the associated negative mental health outcomes compared to their men counterparts [66].

**Table 4. Moderation Effects of Resilience on Depression and Anxiety.**

| Interaction term | Depression | | Anxiety | |
|---|---|---|---|---|
| | Wald Chi-square statistics | p-value | Wald Chi-square statistics | p-value |
| Feel unsafe in public space * Resilience | 0.99 | .319 | 0.35 | .555 |
| Feel unsafe in public transportation * Resilience | 0.09 | .767 | 0.002 | .967 |
| Feel unsafe in neighborhood * Resilience | 1.41 | .235 | 0.81 | .369 |
| Stay away from public spaces * Resilience | 0.02 | .901 | 0.03 | .859 |
| Avoid leaving home * Resilience | 1.00 | .318 | 1.02 | .313 |
| Avoid transportation * Resilience | 0.02 | .887 | 0.08 | .779 |
| Accompany * Resilience | 0.89 | .346 | 3.61 | .057 |

*Note.* Wald chi-square test statistics were for interaction terms from logistic regression models. *p* values are two-tailed. Each row represents a separate model testing the interaction between resilience and the listed safety/behavior variable in predicting depression or anxiety. Resilience was dichotomized into low resilience (<3) and normal to high resilience (≥ [3]).

For Asian American women, the lack of perceived safety in public spaces and neighborhoods is closely linked to adverse mental health outcomes. This finding is well-aligned with the existing evidence demonstrating the critical role of perceived safety of surrounding environments in shaping mental health. Systematic reviews consistently indicate that a lack of perceived safety plays a significant role in determining mental health, even after accounting for other individual and environmental characteristics [20,21]. Given that perceptions of safety are often shaped by the social and built environments, such as neighborhood socioeconomic deprivation and the prevalence of public disorder or crime, it is plausible that women who feel unsafe experience chronic stress due to the constant need for vigilance and fear of harm [67,68]. Such ongoing stress may cause a substantial mental health burden.

We draw on the Person and Environment (P-E) fit theory as a conceptual lens to understand these relationships, which posits that stress and adverse mental health outcomes arise when the environment fails to meet individual needs [69]. In our study, Asian American women's internal need for safety was poorly supported by their external environment. This mismatch can lead to a sense of lack of control and uncertainty, further exacerbating mental health challenges. For example, women may experience heightened intolerance of uncertainty and increased worry, key symptoms of anxiety. Moreover, this group faces a heavy burden of stressors, including race- and gender-based violence and discrimination [6,70], while having limited access to stress-combating resources (e.g., social cohesion, health services) [42,43]. This imbalance may contribute to a heightened sense of vulnerability and adverse mental health outcomes. This finding contributes to the existing literature on the well-established link between neighborhood safety and mental health in racial/ethnic minoritized groups [20,21], offering a more comprehensive understanding of the heavy toll environmental threats—including those in public spaces—take on mental health. Programs and policies that aim to create safe public environments are essential for maintaining mental health well-being in this racial/ethnic minoritized group.

Behavioral modifications taken as coping measures in response to safety concerns, such as avoiding public spaces, staying home, or avoiding public transportation, have a pervasive and significant impact on depression and anxiety. These avoidance behaviors, while often employed as coping mechanisms in the face of stress [22], can exacerbate existing mental health challenges. By isolating individuals from their social networks, reducing social interactions, and limiting access to support resources, these avoidance behaviors can create a vicious cycle of loneliness, depression, and anxiety, as evidenced in a global cross-cultural meta-analysis [27]. Another plausible explanation is that these behavioral modifications may have disrupted the daily routines of Asian American women. As proposed by the Drive to Thrive theory, maintaining daily routines and structures can help individuals develop resilience to cope with stress [71] while disruptions to daily routines can contribute to depression and anxiety, as demonstrated by a recent systematic review and meta-analysis of 53 studies globally on 910,503 individuals [72]. It is also important to consider the bidirectional relationship between safety-related behavioral modifications and mental health, particularly given the cross-sectional nature of this study. Individuals with mood and anxiety disorders may be more likely to rely on avoidance behaviors, especially when under stress [73]. This finding underscores the importance of offering effective resources for Asian American women who cope with stress through these avoidance behaviors. For instance, telehealth could be a valuable tool, allowing access to essential mental health services in an effective and convenient manner [74,75]. Additionally, other online platforms such as Facetime can help them maintain social connections by staying connected with family and friends.

Aligned with the well-established connections between loneliness and mental health [31,32], women in our study who experienced loneliness exhibited higher odds for depression and anxiety compared to those without loneliness. Notably, the prevalence of loneliness in this group of highly educated, English-speaking Asian American women, who tend to have more resources compared to their peers, was strikingly high. In fact, it exceeded the prevalence of loneliness reported in an Asian American elderly sample, most of whom had limited English proficiency [24]. Future research is needed to further investigate the elevated level of loneliness in this group and the contributing factors. Specifically, it is crucial to examine whether these patterns emerged following the surge of anti-Asian hate during the COVID-19 pandemic or if the prevalence was similarly high prior to the pandemic.

The findings on the moderating role of loneliness on the relationship between avoiding public spaces due to safety concerns and anxiety provide nuanced and valuable insights into the complexity of these factors. Notably, Asian American women who did not experience loneliness had higher odds of anxiety when they avoided public spaces. This may suggest that social isolation associated with behavioral modifications—characterized by an objective lack of social interactions with others or the wider community [31]—are linked with greater vulnerabilities to adverse mental health outcomes, compounding the effects of loneliness alone [31,32]. These women might need to maintain active social connectedness through participating in activities that foster interactions with others in the community [76], beyond what is offered within their households. Such interactions could be crucial for them to obtain social support and a sense of belonging. For these women, withdrawing from public spaces could limit their access to these critical support systems, disrupting their coping strategies and heightening feelings of isolation, which, in turn, associated with greater odds of anxiety [31,32].

Conversely, for women who were already lonely, the relationship between avoiding public spaces and anxiety was not significant. This may be because loneliness itself has already profoundly impacted their anxiety levels, overshadowing the effects of further social withdrawal. Additionally, women who were lonely may have fewer social connections or more restricted social networks to begin with [35,36], meaning that avoiding public spaces does not represent a significant additional loss of social interaction or support, and thus has a less pronounced effect on their anxiety. Future research should continue to explore the moderating roles of loneliness and clarify other potential confounding factors, such as social networks and social support, to inform the development of effective tailored mental health interventions based on loneliness status.

This study reveals the significant associations between resilience and mental health outcomes, as women with low resilience had higher odds of experiencing depression and anxiety compared to those with moderate to high resilience. This is consistent with well-documented evidence linking resilience with favorable mental health outcomes across diverse populations [39–41]. It is essential to acknowledge the resilience demonstrated by this population in the face of numerous challenges. However, over 40% of the women reported low resilience, challenging the common narrative of Asian Americans as inherently resilient individuals who maintain positive beliefs about adversity and are able to thrive despite challenges [77,78]. Resilience was not found to significantly moderate the associations between perceived safety or safety-related behavioral modifications and either depression or anxiety. This was not surprising from a socioecological perspective: while individual resilience may contribute to better overall mental health, it may not be sufficient to counterbalance the negative associations of accumulated environmental challenges [79], as reflected by their lack of perceived safety in public areas and related behavioral modifications, on mental health. Furthermore, the potential benefits of individual resilience in promoting mental health may be constrained by the availability and accessibility of the socioecological resources in the environment, such as physical and social capital [42,43]. The lack of perceived safety could reflect a deficiency in these resources, and behavioral modifications through avoidance might further hinder access to them. Thus, resilience alone may not be sufficient in mitigating mental health challenges without concurrent efforts to address community and structural determinants.

A growing number of programs are being developed to address the critical issue of women's safety. One widely implemented initiative in the US is the Sexual Assault Response Teams (SARTs) program, which enhances community responses to sexual assault by promoting collaboration among key stakeholders across medical, criminal justice, and mental health/advocacy systems, thereby improving women's help-seeking experiences [80]. Technology is also being leveraged, such as the "bSafe" mobile app, which enables women to share their real-time GPS location with friends and family, schedule check-in times, and use a "walk with you" option to help ensure they arrive at their destination safely [81]. These programs could be a promising start for culturally tailoring and testing their effectiveness among Asian American women.

This study is not without limitations. First, the cross-sectional design restricts our ability to infer causality. For instance, resilience is a dynamic process [82–84] in which coping with stressors can help women cultivate new capacities and

competencies, potentially offering partial immunization against future stressors and preventing adverse mental health outcomes [85]. However, this dynamic aspect was not captured by our current study design. Second, although we successfully recruited women from various Asian groups and East Asians were more often the victims of violence since COVID-19 [86], certain ethnic groups, particularly South Asian populations (e.g., Pakistani, Nepali and Bangladeshi), are underrepresented in our sample. The small sample size in these groups prevents us from conducting disaggregated analyses by specific Asian groups. We conducted exploratory analyses testing interactions between Asian subgroup membership and key independent variables; most were non-significant, suggesting that associations were generally consistent across groups (not presented in the results). Future studies should include larger and more diverse samples to allow for robust subgroup analyses. Third, the use of convenience sampling and related selection bias limits the representativeness of our sample. For instance, most women in our sample had college-level education and were proficient in English, a group likely having more resources than peers with lower educational attainment or limited English proficiency. This limits generalizability, though it is notable that our findings still counter the model minority stereotype by showing that even highly educated and English-proficient Asian American women report substantial mental health burdens. Lastly, we employed a single item to measure perceived safety of the public space, transportation, and neighborhood respectively, which may not adequately capture the multidimensional nature of each construct such as the built environment and the social disorders of neighborhoods.

## Conclusion

Our study significantly contributes to the broader literature on the mental health of racial/ethnic minority groups by illustrating the heavy toll of environmental threats, as indicated by perceived safety and safety-related behavioral modifications, on mental health. The findings also reinforce the need for targeted mental health interventions that address the unique safety challenges faced by Asian American women. Enhancing public safety through research, program, and policy efforts are critical for mitigating mental health challenges in this population. Future research should continue to explore these relationships longitudinally and across diverse conceptualizations of resilience to develop more effective, culturally relevant interventions that address the specific needs of Asian American women.

## Acknowledgments

We are grateful to our participants who generously shared their experiences with us.

## Author contributions

**Conceptualization:** Jiepin Cao, Sarah Shevrin, Sahnah Lim.

**Formal analysis:** Jiepin Cao.

**Writing – original draft:** Jiepin Cao.

**Writing – review & editing:** Jiepin Cao, Sarah Shevrin, Linh M. An, Jennifer A. Wong, Sugy Choi, Stella S. Yi, Chau Trinh-Shevrin, Sahnah Lim.

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
