## [Decision Letter · Decision Letter 0]

31 Jul 2025

Dear Dr. Cao,

Thank you for submitting your manuscript to PLOS ONE. After careful consideration, we feel that it has merit but does not fully meet PLOS ONE’s publication criteria as it currently stands. Therefore, we invite you to submit a revised version of the manuscript that addresses the points raised during the review process.

https://journals.plos.org/plosone/s/submission-guidelines#loc-laboratory-protocols . Additionally, PLOS ONE offers an option for publishing peer-reviewed Lab Protocol articles, which describe protocols hosted on protocols.io. Read more information on sharing protocols at https://plos.org/protocols?utm_medium=editorial-email&utm_source=authorletters&utm_campaign=protocols .

We look forward to receiving your revised manuscript.

Kind regards,

Renata Komalasari

Academic Editor

PLOS ONE

Journal Requirements:

“This work was supported by the National Institutes of Health (NIH) – National Institute on Minority Health and Health Disparities [grant number U54MD000538] and by the NIH – National Heart, Lung, and Blood Institute [grant number 1R01HL160324].”

“This work was supported by the National Institutes of Health (NIH) – National Institute on Minority Health and Health Disparities [grant number U54MD000538] and by the NIH – National Heart, Lung, and Blood Institute [grant number 1R01HL160324].”

Additional Editor Comments:

Thank you for the opportunity to handle this manuscript. I have read the text with interest and have some questions for you and issues that I would like you to address:

Please provide coding for categorical variables to help with interpretation of logistic regression results.

Lines 271 - 273: The authors cited an aOR = 4.77, 95% CI: 1.24, 18.34, showing that among women who did not feel lonely, those who stayed away from public spaces due to safety concerns had 4.77 times the odds of reporting anxiety compared to those who did not stay away. Are you referring to aOR value of 4.81 as reported in in Table 3? Is yes, which value is correct: 4.77 or 4.81?

Lines 273-275 Is this statement refers to the simple slope analysis interpretation. If yes, I suggest indicating this in the sentence. The authors stated that among women who were lonely, staying away from public spaces was associated with lower odds of reported anxiety compared to those who did not stay away. The value of aOR = 0.93 was cited but I did not see such value in Table 3 as the authors indicated. However, learning from simple slope graph in Fig 1, my thought is that among those who were lonely there was no differences in anxiety reports between those who stayed away from public spaces or not. Hence, no moderating effect was observed amongst women who were lonely. Therefore, the sentence "Loneliness moderated the association between avoiding public spaces and anxiety" found in the abstract (line 37) and in line 271 may be misleading as moderation effect of loneliness was observed only among women who were not lonely. Please ensure correct interpretation of statistical results across the paper.

Please follow APA format in reporting statistical reports: for example: please use small letter 'p' in reporting p value, instead of capital P.

Reviewers' comments:

Reviewer's Responses to Questions

**Comments to the Author**

1. Is the manuscript technically sound, and do the data support the conclusions?

Reviewer #1: Yes

Reviewer #2: Partly

2. Has the statistical analysis been performed appropriately and rigorously?

Reviewer #1: Yes

Reviewer #2: I Don't Know

3. Have the authors made all data underlying the findings in their manuscript fully available?

Reviewer #1: No

Reviewer #2: No

4. Is the manuscript presented in an intelligible fashion and written in standard English?

Reviewer #1: Yes

Reviewer #2: Yes

Reviewer #1: Introduction

1. In lines 48-49, “Existing literature suggests that this group, particularly Asian American women, face a disproportionate burden of mental health challenges.”

If you're treating "group" as singular, you should use "faces" instead of “face”.

2. In lines 57-58, “This largely stems from their marginalization based on intersectional identities as both Asian and women, rooted in a legacy of gendered racism.”

To strengthen the statement’s credibility, appropriate citations and references should be included.

3. In lines 106-108, “However, the influence of loneliness on the mental health of Asian American women, particularly in the context of environmental safety threats remains largely unexplored.”

Your paragraph is generally clear. Reorganizing the sentences can improve logical flow. You may want to move this research gap sentence after the current research scope regarding the general Asian American community.

Methods

4. In lines 150-152, “REDCap is a secure, HIPAA (Health Insurance Portability and Accountability Act) compliant software platform designed to support data collection for research studies.”

It is important to cite the official description so that readers can understand its origin and validation

Measures

5. Even though you mentioned the reliability and validity of some measurements, it is still important to check and report the internal consistency (e.g., Cronbach’s alpha) of each measurement for your sample.

Result

6. In Line 245, “Table 1. Sample Characteristics of Asian American Women (n=349)”

Please follow APA 7th edition guidelines for table headings and formatting and avoid using Excel-style tables. (Apply these suggestions for other tables as well.)

One example is below:

“

Table 1.

Sample Characteristics of Asian American Women (n=349) “

7. In lines 269 to 271:

The statement “Loneliness did not moderate the relationship between any avoidant behaviors and depression. Loneliness moderated the relationship between staying away from public spaces and anxiety” lacks clear statistical evidence to support these claims. I recommend including the corresponding p-values or related statistical measures in the text to substantiate these conclusions.

8. In lines 272 to 276:

Some adjusted odds ratios, such as aOR = 4.77, do not appear in Table 2 or Table 3. Please clarify which table or figure they are presented in and ensure that the numbering and explanations in the text align with the corresponding tables or figures. If these adjusted odds ratios are illustrated in Figure 1, please label them clearly to improve clarity for readers.

There is a mention of Figure 1 related to the interaction effects on anxiety, but it is not explained clearly. Please ensure that Figure 1 is clearly presented, with appropriate labeling of any significant numbers, and properly referenced in the text.

9. In lines 285:

In the phrase “(all P>0.05),” the “P” should be italicized for consistency with statistical reporting conventions.

Discussion/Conclusion

10. In line 328 – Theoretical Background:

You refer to the Person and Environment (P-E) fit theory in line 328, but it isn’t integrated into your research questions. Instead of addressing P-E fit theory in the discussion, I recommend adding a brief description of the P-E fit theory in the literature review or theoretical framework section to help guide the reader in understanding how this theory informs your study.

Reviewer #2: SUMMARY: In the wake of a rise in anti-Asian violence in the United States, the present study explores potential impact of perceived safety (across three types of locations) and behavioral changes due to safety concerns (as independent variables) upon depression and anxiety (as dependent variables) for Asian American women, with self-reported resilience and loneliness explored as potential moderating variables. From a convenience sample of n-345 Asian American Women, findings support a significant positive correlations between most of the independent variables and depression, anxiety, or both. The moderating variable of loneliness is only found to have significant influence on anxiety for respondents who stay away from public spaces (a behavioral change due to safety concern). Resilience is not found to have any significant influence on depression or anxiety. Authors elaborate on their methods, theoretical contexts, and potential interventions from these findings.

1. The manuscript is technically adequate for reporting the study as designed, yet demonstrates methodological weaknesses that go as far back as the design and analysis stages. I recommend the authors start over with more rigorous and informed analysis.

My greatest concern is that even though disaggregated demographic data has been collected, it is mentioned late and infrequently, then omitted from analysis without explanation or justification. Asian American women are not a monolith (I am thinking here of decolonial work by Rosalind S. Chou, as well as Aihwa Ong's "Cultural Citizenship as Subject-Making"); while a case could be made that perceptions of safety are more linked to how one is stereotyped in the world than one's own experience, this paper does not make any such case. There is no indication that demographics (including but not limited to ancestral nationality) were given much consideration at all during analysis; readers are given inadequate demographic descriptives and then told they were omitted from analysis due to lack of significance. The authors also seem to be unaware of the terms or value of convenience sampling or weighted data (cf. lines 435-441), so their snowball sample is not compared to any broader population figures at all. As a convenience sample, my previous point is all the more glaring: generalizability was never going to be strong, so the quality of the data lies in relationships that are largely unexplored here. No effort is made to justify the omission, let alone explore possible interactions with other findings. Without a more thorough report and contextualization of sample demographics -- ancestry and time in the U.S. most urgently -- readers can hardly surmise whether further items (such as regional and urban/suburban/rural distribution) might be missing.

Perhaps most troubling is the authors' murky claims of causality. Measures are introduced and somewhat described, but no grounds are given for the causal relationship as examined -- in fact, the Discussion once implies an inverted causality (lines 356-359) and elsewhere implies no causality exists at all (line 428). No statistics or sources are given for potential overlap between depression and anxiety, nor between resilience and loneliness.

One more fundamental flaw: authors imply (lines 193-195) that the survey was cut short for respondents who reported feeling safe, and this lost data undermines the analysis; these responses could provide a vital control for internal and external comparison, and without them the data is incomplete for whatever portion (unreported) describe no unsafe feelings. (That descriptive statistics are not reported fully enough for me to tell if my reading is accurate is also an issue.)

I have secondary concerns as well.

The authors adequately contextualize their research question amid a rise of anti-Asian violence, but do not mention whether things may have changed as the public conscience has moved away from COVID-19. If such data does not exist yet, so be it, but it warrants mention, especially in Discussion. Neither is there adequate elaboration about why the sample includes all Asian Americans instead of only those of East Asian ancestry, even though the distinction is raised on lines 239-240 and, to the best of my knowledge, highly relevant to COVID-era hate crimes.

There is a recurring lack of clarity as to when authors have chosen a tool, cutoff, or theoretical tangent because it has been recommended in other research versus their own whims and convenience. There is not necessarily a wrong answer, but authors must give one; examples include why the cutoff for depression measurement is different from the cutoff for anxiety measurement, the use of an unspecified "pre-selected list" (mentioned in lines 222-224 but notably not during Methods), and two theoretical frameworks introduced in discussion with neither buildup nor application. Many times, a comparison is made (examples include "high prevalence", line 370, or "disproportionate", line 334, but there are many others) without specifying who is being compared: general population, American women in general, Asian Americans in general, existing research of similar samples, or other groupings within this study? Authors must be explicit every time.

2. I think a lot more descriptives should be included, as detailed above, and there are technical flaws in how several statistics are reported, but the statistics themselves seem fine as presented.

3. The statistics were not made available to reviewers for examination.

4. The manuscript is rough and would warrant a complete outside copyedit; the most egregious errors are inconsistency of own terms (e.g. "perceived safety" vs. "safety concerns", "behavioral modifications" vs. "modified behaviors") and capitalization that defies statistical convention (for example, "p-value" should never be capitalized because authors are discussing a sample -- capital letters reflect population-level data), but two paragraphs (lines 204-212 and lines 290-302) are sufficiently convoluted and unprofessional as to undermine the study itself. The authors seem to lack both technical knowledge (such as how to insert special characters like "≥" in Table 1, Education) and sensitivity to not "other" respondents whose responses were not listed (Table 1, Sexual Orientation). My preference that the analysis address mixed race and transgender identities is, admittedly, unlikely to provide statistically significant findings, but the present items still fail to meet minimal standards because almost none of the demographic items collected specify how they were structured in the survey: were respondents given a list, a blank space, or a combination of the two? Several times, figures cited from other studies fail to specify whether they report a similar population (Asian American women) or an adjacent one (Asian Americans at large, women in the U.S., etc.).

The Discussion is deeply flawed, introducing almost as many new concepts as the introduction and making assertions that overstate or outright contradict the findings as reported (lines 394-395 are probably the worst). Several of the finest passages in the text (lines 321-323, 337-341, 366-369, and 382-387) are misplaced here and buried between passages of unscientific commentary.

I applaud the authors for their hard work, their successful research grant, and their data collection. I hope this feedback leads to strong works in the future.

**Do you want your identity to be public for this peer review?** For information about this choice, including consent withdrawal, please see our Privacy Policy

Reviewer #1: **Yes: ** Rongfang Zhan

Reviewer #2: No

---

## [Author Response · Author response to Decision Letter 1]

17 Sep 2025

Dear Dr. Renata Komalasari,

We appreciate the opportunity to revise and resubmit our manuscript “Perceived Safety and Mental Health among Asian American Women: Exploring the Moderating Role of Loneliness and Resilience” to PLOS ONE. Please find our point-by-point responses below. The comments of editor and reviewers are indicated, and our responses immediately follow. Please also find a revised manuscript with tracked changes and a cleaned manuscript.

Comments from editor and reviewers

Journal Requirements:

• Response: We have followed the PLOS ONE style requirement and ensured that our manuscript, including file naming and formatting, now fully conforms to the journal’s guidelines.

“This work was supported by the National Institutes of Health (NIH) – National Institute on Minority Health and Health Disparities [grant number U54MD000538] and by the NIH – National Heart, Lung, and Blood Institute [grant number 1R01HL160324].”

• Response: Thank you and we have included the following amended Funding Statement to the cover letter: “This work was supported by the National Institutes of Health (NIH) – National Institute on Minority Health and Health Disparities [grant number U54MD000538],National Heart, Lung, and Blood Institute [grant number 1R01HL160324], and National Institute of Mental Health [grant number R34MH136914-01A1]. There was no additional external funding received for this study. The funders had no role in study design, data collection and analysis, decision to publish, or preparation of the manuscript.”

“This work was supported by the National Institutes of Health (NIH) – National Institute on Minority Health and Health Disparities [grant number U54MD000538] and by the NIH – National Heart, Lung, and Blood Institute [grant number 1R01HL160324].”

• Response: We have clarified that the funders were not involved in study design, data collection and analysis, decision to publish, or manuscript preparation in the cover letter. We have included the following amended Funding Statement to the cover letter: “This work was supported by the National Institutes of Health (NIH) – National Institute on Minority Health and Health Disparities [grant number U54MD000538],National Heart, Lung, and Blood Institute [grant number 1R01HL160324], and National Institute of Mental Health [grant number R34MH136914-01A1]. There was no additional external funding received for this study. The funders had no role in study design, data collection and analysis, decision to publish, or preparation of the manuscript.”

• Response: We have updated the Data Availability statement to include an institutional contact for data access. NYULH IRB Operations: IRB-Info@NYULangone.org

• Response: Thank you!

• Response: We have carefully reviewed our reference list to ensure accuracy. In response to reviewer concerns, we have added several new references to strengthen the clarity and rigor of the manuscript; these additions are detailed and justified in our responses to specific reviewer comments. None of the references cited in our manuscript have been retracted.

Additional Editor Comments:

Thank you for the opportunity to handle this manuscript. I have read the text with interest and have some questions for you and issues that I would like you to address:

1. Please provide coding for categorical variables to help with interpretation of logistic regression results.

• Response: Thank you and we have now specified how categorical variables were coded in the measures section. See updated line 186-187; 194-195; 202-203; 213-214; 220; 228-229.

2. Lines 271 - 273: The authors cited an aOR = 4.77, 95% CI: 1.24, 18.34, showing that among women who did not feel lonely, those who stayed away from public spaces due to safety concerns had 4.77 times the odds of reporting anxiety compared to those who did not stay away. Are you referring to aOR value of 4.81 as reported in in Table 3? Is yes, which value is correct: 4.77 or 4.81?

• Response: Thank you for pointing this out. To clarify, the value of 4.81 reported in Table 3 is the Wald chi-square statistic for the interaction term (stay away from public spaces × loneliness) with the corresponding p value of 0.028, which was presented to indicate statistical significance of the interaction. The aOR = 4.77, 95% CI: 1.24–18.34 represents the conditional effect of staying away from public spaces on anxiety among women who did not report loneliness. These conditional effect estimates are not displayed in Table 3 but are reported in-text to illustrate how the moderation effect of loneliness operates. We have revised the Results section to explicitly note this distinction for clarity.

3. Lines 273-275 Is this statement refers to the simple slope analysis interpretation. If yes, I suggest indicating this in the sentence. The authors stated that among women who were lonely, staying away from public spaces was associated with lower odds of reported anxiety compared to those who did not stay away. The value of aOR = 0.93 was cited but I did not see such value in Table 3 as the authors indicated. However, learning from simple slope graph in Fig 1, my thought is that among those who were lonely there was no differences in anxiety reports between those who stayed away from public spaces or not. Hence, no moderating effect was observed amongst women who were lonely. Therefore, the sentence "Loneliness moderated the association between avoiding public spaces and anxiety" found in the abstract (line 37) and in line 271 may be misleading as moderation effect of loneliness was observed only among women who were not lonely. Please ensure correct interpretation of statistical results across the paper.

• Response: Thank you for this thoughtful comment. We have clarified in the revised manuscript that we are reporting results from a simple slopes (conditional effects) analysis. Specifically, the estimates (aOR = 4.77, 95% CI: 1.24–18.34; aOR = 0.93, 95% CI: 0.53–1.64) are conditional effect estimates that were not presented in Table 3, which instead reports the Wald chi-square test for the interaction terms.

• We agree that the figure suggests no significant differences among women who reported loneliness, consistent with the conditional estimate (aOR = 0.93, 95% CI: 0.53–1.64, not statistically significant). By definition, moderation is present when the association between an independent variable and an outcome differs across levels of a moderator. In our study, the effect of staying away from public spaces on anxiety differed by loneliness status: it was significant among women who were not lonely but not significant among those who were lonely. We have now revised the Results and abstract to clearly explain the nature of the moderation effect of loneliness.

4. Please follow APA format in reporting statistical reports: for example: please use small letter 'p' in reporting p value, instead of capital P.

• Response: Thank you and we have updated the statistical reporting to align with APA 7th formatting, including non-capitalized and italicized p value.

Reviewers' comments:

Review Comments to the Author

Reviewer #1: Introduction

1. In lines 48-49, “Existing literature suggests that this group, particularly Asian American women, face a disproportionate burden of mental health challenges.”

If you're treating "group" as singular, you should use "faces" instead of “face”.

• Response: Thanks so much for noting this. We have changed “face” to “faces”.

2. In lines 57-58, “This largely stems from their marginalization based on intersectional identities as both Asian and women, rooted in a legacy of gendered racism.”

To strengthen the statement’s credibility, appropriate citations and references should be included.

• Response: Thank you and we have added citation to support our statement.

• Forbes N, Yang LC, Lim S. Intersectional discrimination and its impact on Asian American women’s mental health: A mixed-methods scoping review. Front Public Health [Internet]. 2023. Available from: https://www.frontiersin.org/articles/10.3389/fpubh.2023.993396

3. In lines 106-108, “However, the influence of loneliness on the mental health of Asian American women, particularly in the context of environmental safety threats remains largely unexplored.”

Your paragraph is generally clear. Reorganizing the sentences can improve logical flow. You may want to move this research gap sentence after the current research scope regarding the general Asian American community.

• Response: Thank you, we have now moved this sentence to follow the discussion of existing evidence on the general Asian American community to enhance the logical flow.

Methods

4. In lines 150-152, “REDCap is a secure, HIPAA (Health Insurance Portability and Accountability Act) compliant software platform designed to support data collection for research studies.”

It is important to cite the official description so that readers can understand its origin and validation

• Response: Thank you for this suggestion. We have added the official REDCap references to acknowledge the origin of the platform: “REDCap is a secure, HIPAA (Health Insurance Portability and Accountability Act) compliant software platform designed to support data collection for research studies (47,48)”

47. Harris PA, Taylor R, Minor BL, Elliott V, Fernandez M, O’Neal L, et al. The REDCap consortium: Building an international community of software platform partners. J Biomed Inform. 2019;95:103208.

48. Harris PA, Taylor R, Thielke R, Payne J, Gonzalez N, Conde JG. Research electronic data capture (REDCap)—A metadata-driven methodology and workflow process for providing translational research informatics support. J Biomed Inform. 2009;42(2):377–81.

Measures

5. Even though you mentioned the reliability and validity of some measurements, it is still important to check and report the internal consistency (e.g., Cronbach’s alpha) of each measurement for your sample.

• Response: Thank you for this helpful suggestion. We have now reported the internal consistency (Cronbach’s α) for each measurement scale within our sample in the Measures section, which also complements the psychometric information on the scales. The UCLA Loneliness Scale (3-item), The Brief Resilience Scale, PHQ-9, and GAD-7 all demonstrated good to excellent internal consistency (αs = .82–.91).

Result

6. In Line 245, “Table 1. Sample Characteristics of Asian American Women (n=349)”

Please follow APA 7th edition guidelines for table headings and formatting and avoid using Excel-style tables. (Apply these suggestions for other tables as well.)

One example is below:

“Table 1. Sample Characteristics of Asian American Women (n=349) “

• Response: Thank you for noting this. We reviewed the PLOS ONE formatting guidelines, which specifically require that tables be created as cell-based tables in Microsoft Word or embedded from Microsoft Excel. Because of these journal requirements, we did not modify the table formatting: https://journals.plos.org/plosone/s/file?id=wjVg/PLOSOne_formatting_sample_main_body.pdf. However, within these journal formatting constraints, we ensured that table titles and notes adhere to APA 7th edition style conventions whenever possible.

7. In lines 269 to 271:

The statement “Loneliness did not moderate the relationship between any avoidant behaviors and depression. Loneliness moderated the relationship between staying away from public spaces and anxiety” lacks clear statistical evidence to support these claims. I recommend including the corresponding p-values or related statistical measures in the text to substantiate these conclusions.

• Response: Thank you for noting this. We have clarified the statistical evidence supporting our conclusions: “Loneliness did not moderate the relationsh

---

## [Decision Letter · Decision Letter 1]

4 Nov 2025

Perceived Safety and Mental Health among Asian American Women: Exploring the Moderating Role of Loneliness and Resilience

PONE-D-24-59672R1

Dear Dr. Cao,

We’re pleased to inform you that your manuscript has been judged scientifically suitable for publication and will be formally accepted for publication once it meets all outstanding technical requirements.

Kind regards,

Renata Komalasari

Academic Editor

PLOS ONE

Additional Editor Comments (optional):

Other than few minor suggestions from the reviewer, I think the paper has improved and can be published.

Reviewers' comments:

Reviewer's Responses to Questions

**Comments to the Author**

Reviewer #1: All comments have been addressed

2. Is the manuscript technically sound, and do the data support the conclusions?

Reviewer #1: Yes

3. Has the statistical analysis been performed appropriately and rigorously?

Reviewer #1: Yes

4. Have the authors made all data underlying the findings in their manuscript fully available?

Reviewer #1: Yes

5. Is the manuscript presented in an intelligible fashion and written in standard English?

Reviewer #1: Yes

Reviewer #1: 1. The statement in line 26: “This study aimed to 1) examine the relationship between perceived safety with mental health outcomes (i.e., depression and anxiety).

Check grammar does the sentence should use “between…..and” or “between….with” in line 26.

2. In abstract, author should briefly mention what statistics method were used in this study.

3. Line 44-45: I did not see keywords after the abstract section.

4. Check line 61-63: This sentence is not clear, please clarify: “35%, 37%, and 40% reported sexual harassment.”

5. The perceived safety definition was defined in the abstract, but in the introduction. It would be better if the authors could revisit the definition of perceived safety in the introduction or background sections. [line 78 section]

6. Check the grammar in line 152: “ examine the relationships between perceived safety in public spaces, public transportation, and neighborhoods with mental health outcomes (i.e., depression and anxiety).

7. Check line 281: Please consider using a small letter in “See” Table 1. Please check APA style in formatting tables 1, 2, and 3.

check the same format in table 2 , table 3…

9. Line 358 In your response, you mentioned using the P-E fit theory only as an interpretative lens. However, in line 358, once again, you mentioned it as a conceptual lens, which may be confusing to readers. Please clarify.

no

**Do you want your identity to be public for this peer review?** For information about this choice, including consent withdrawal, please see our Privacy Policy

Reviewer #1: **Yes: ** Rongfang Zhan

---

## [Editor Report · Acceptance letter]

PONE-D-24-59672R1

PLOS ONE

Dear Dr. Cao,

I'm pleased to inform you that your manuscript has been deemed suitable for publication in PLOS ONE. Congratulations! Your manuscript is now being handed over to our production team.

Kind regards,

on behalf of

Dr. Renata Komalasari

Academic Editor

PLOS ONE